# In-hospital survival and predictors of mortality among stroke patients at a tertiary hospital in Ghana: A retrospective cohort study

**Sulemana Baba Abdulai**[1]*, **Julius Kwabena Karikari**[1,2], **Penias Tembo**[3,4], **Theogene Habumugisha**[5], **William Tembo**[6], **Alhaji Ibrahim Cobbinah**[1], **Yeetey Enuameh**[1]

**1** Department of Epidemiology and Biostatistics, School of Public Health, Kwame Nkrumah University of Science and Technology, Kumasi, Ghana, **2** University Hospital, Kwame Nkrumah University of Science and Technology, Kumasi, Ghana, **3** Department of Epidemiology and Biostatistics, Arnold School of Public Health, University of South Carolina, Columbia South Carolina, United States of America, **4** Cancer Prevention and Control Program, University of South Carolina, Columbia South Carolina, United States of America, **5** Department of Global Public Health and Primary Care, Center for International Health, University of Bergen, Bergen, Norway, **6** Department of Internal Medicine, Neurology Division, University Teaching Hospital, Lusaka, Zambia

\* babasulemanaabdulai@gmail.com

## Abstract

### Introduction

Stroke is a leading cause of death and disability worldwide. In Ghana, national estimates show a prevalence of 7.9% and an incidence rate of 1.2%, placing a significant burden on the health system. This study aimed to estimate in-hospital survival rates and identify predictors of mortality among stroke patients admitted to Tamale Teaching Hospital (TTH).

### Methods

A retrospective electronic medical records review was conducted using data from the Lightwave Health Information Management System (LHIMS) and patients' registry from January 1 2021 to December 31 2023. Kaplan-Meier survival curve was used to determine the survival rate of stroke patients (mean follow-up: 67 days). Cox proportional hazard regression determined the association between risk factors and survival time. Crude and adjusted hazard ratios with 95% confidence intervals were presented. A p-value of 0.05 was considered statistically significant.

### Results

A total of 998 stroke patients were included, of which 39.4% died. The overall survival rate was 21% at the end of follow-up (182 days). The survival probability for female stroke patients was lower compared to males. Female sex (AHR = 1.33, 95% CI: 1.07–1.65), Dagomba ethnicity (AHR = 1.31, 95% CI: 1.05–1.62), pneumonia

**Data availability statement:** All relevant data are within the manuscript and its Supporting Information files.

**Funding:** The author(s) received no specific funding for this work.

**Competing interests:** The authors have declared that no competing interests exist.

(AHR = 1.59, 95% CI: 1.27–1.97), diabetes mellitus (AHR = 0.62, 95% CI: 0.47–0.82), systolic blood pressure ≥ 130 mmHg (AHR = 1.35, 95% CI: 1.06–1.73), and temperature ≥ 37.5°C (AHR = 1.79, 95% CI: 1.37–2.33) were predictors of mortality.

## Conclusion

Female stroke patients experienced higher mortality and lower survival compared to males. The identified mortality predictors underscore the importance of focused interventions to enhance survival outcomes at TTH. Given the retrospective design and possible unmeasured confounders, the lower mortality risk among diabetic patients should be interpreted with caution, and prospective studies are warranted to confirm these associations.

## Introduction

Stroke is an acute neurological dysfunction of the brain, spinal cord, or retina caused by focal ischemia (infarction) or haemorrhage, including cases due to cerebral venous thrombosis [1]. It is a multifaceted disease influenced by risk factors such as hypertension, diabetes, and heart failure [2]. Globally, stroke remains a leading cause of death and disability, responsible for significant health burdens, including an annual loss of approximately 1,484 disability-adjusted life years (DALYs) per 100,000 people [3]. Survivors often endure long-term physical, cognitive, and emotional challenges, imposing substantial demands on families, social, and healthcare systems [4].

In sub-Saharan Africa (SSA), the burden of stroke is rapidly increasing due to an epidemiological shift towards non-communicable diseases (NCDs) [5]. Stroke incidence and mortality rates in SSA are alarmingly high, with incidence reaching up to 1,460 cases per 100,000 people, and a three-year fatality rate exceeding 80% [6]. Challenges such as inadequate health infrastructure, limited neurologists, and low awareness of stroke symptoms exacerbate poor outcomes in the region [7]. In Ghana, stroke ranks among the top five causes of death, with concerning trends in rising admissions and a one-month hospital fatality rate of up to 43% [8,9]. Over the past three decades, the country has witnessed a consistent rise in stroke incidence, admissions, and mortality. For instance, stroke-related admissions in Kumasi increased from 5.32 per 1,000 in 1983 to 13.85 per 1,000 in 2010, with one-month fatality rates reaching 41% in the Central Region [9]. Key modifiable risk factors, including high salt intake, physical inactivity, and limited consumption of green leafy vegetables, are prevalent in Ghana [10]. Additionally, younger populations under 40 years are increasingly affected, posing significant socio-economic challenges [11].

The Tamale Teaching Hospital (TTH) in Northern Ghana, a major referral centre, reports high stroke-related mortality, with stroke ranking as the third leading cause of death in its Accident and Emergency Department [9]. Factors such as delayed presentation, socio-cultural barriers to healthcare, and resource constraints contribute to poor stroke outcomes [9]. While stroke-related mortality has been extensively studied, limited data exist on survival and its determinants, particularly in the Ghana context [7,12].

Despite significant advancements in stroke care, limited region-specific data on stroke survival and its determinants impede the development of targeted interventions in Ghana. Current research has predominantly focused on stroke mortality, with few studies on survival outcomes and their predictors [12–14]. This knowledge gap hinders the development of evidence-based stroke management strategies in LMICs, including the establishment of dedicated stroke units, the availability of diagnostic tools such as MRI machines, and the training of healthcare professionals essential for improving stroke care [15–17]. This study aims to address these gaps by investigating the in-hospital survival rate and predictors of mortality among stroke patients admitted to the Tamale Teaching Hospital, Ghana. The availability of evidence on the determinants of survival among stroke patients will help to inform healthcare planning, enhance resource allocation, and support the development of data-driven public health policies in Ghana. Furthermore, the study aligns with the WHO's Agenda 2030 for Sustainable Development, particularly its goal of reducing mortality from major NCDs by one-third by 2030.

## Materials and methods

### Study design

This study employed a retrospective cohort design using routinely collected hospital data. Stroke patients admitted to Tamale Teaching Hospital between 2021 and 2023 were followed from admission until discharge or in-hospital death. The retrospective cohort design was appropriate for identifying predictors of in-hospital mortality among stroke patients using existing clinical records.

### Patient identification

We conducted a retrospective review of electronic medical records using the Lightwave Health Information Management System (LHIMS) to identify all patients admitted with a diagnosis of stroke at the Tamale Teaching Hospital (TTH) between January 1, 2021, and December 31, 2023. Medical records were accessed for research purposes between April 15 and June 15, 2024. The study included patients aged 18 years and above who were diagnosed with either ischemic or hemorrhagic stroke during the study period. No a priori sample size or power calculation was performed because the study included all eligible stroke cases within the 3 years. Patients were excluded if their medical records were incomplete or missing, or if the type of stroke was not specified as either ischemic or hemorrhagic (Fig 1). The authors had no access to information that could identify individual participants at any point during or after data collection. All data used for the study were fully anonymised prior to analysis to ensure the privacy and confidentiality of patients. The study adhered to the principles of the Declaration of Helsinki for research involving human participants. Ethical approval for the study was obtained from the Committee on Human Research, Publications and Ethics at the Kwame Nkrumah University of Science and Technology (KNUST), under approval number CHRPE/AP/237/24. Given the retrospective nature of the study, the requirement for informed consent was waived.

### Study setting

The study was centred at the Tamale Teaching Hospital (TTH) in Tamale, Northern region. The hospital is a referral centre for all other hospitals within the five regions, including the Northeast, Upper East, Savanna, and Upper West [9]. The hospital serves an approximate population of over 4 million people [18]. It is currently Ghana's third-largest tertiary specialised healthcare facility [19]. In addition to providing medical care to patients, the hospital acts as a teaching facility for the University for Development Studies and many nursing colleges in the region [9]. At the time of the study, stroke care was provided within the general medical wards, as the hospital did not have a dedicated stroke unit. Stroke patients were managed by a multidisciplinary team comprising specialist physicians (including an endocrinologist and a gastroenterologist), medical officers, nurses, pharmacists, and physiotherapists. As a high-volume referral facility, TTH manages significant numbers of stroke patients; for example, 105 ischemic stroke admissions were recorded between January and October 2021 [9].

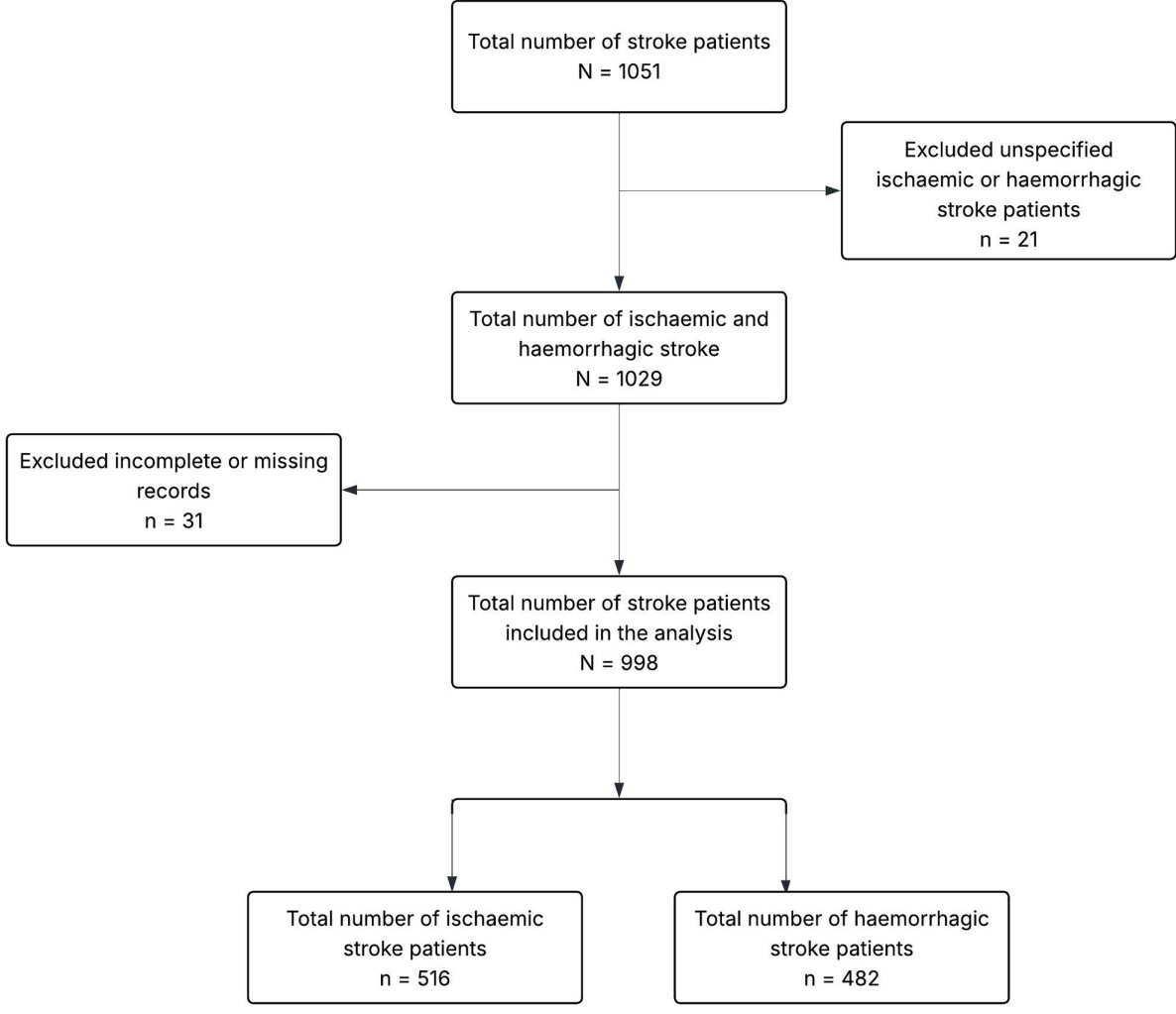

**Fig 1. Identification and eligibility criteria used for the study population in the analysis.**

## Study variables

The event of interest was in-hospital death. Survival time was defined as the number of days from admission to death, with discharge alive treated as a censoring event. No post-discharge follow-up data were available. Predictors were grouped into demographic/baseline characteristics (age, sex, educational level, occupation, ethnicity, family history of stroke, smoking status, alcohol intake, body temperature, systolic and diastolic blood pressure, heart rate, respiratory rate, and Glasgow Coma Score) and clinical characteristics (family medical history, hypertension, diabetes mellitus, kidney disease, pneumonia, stroke type, atrial fibrillation, right and left upper limb weakness, seizures, aphasia, dysarthria, and dysphagia) (**Table 1**). Dates of admission and discharge or death were collected to compute time to in-hospital death or censoring.

## Data sources and tools

Data for this study were extracted from the Lightwave Health Information Management System (LHIMS) and patient registers at the Tamale Teaching Hospital (TTH) for the years 2021–2023.

**Table 1. Demographic/baseline and clinical characteristics of stroke patients at TTH.**

| Characteristics | Frequency (N = 998) | Percentage (%) |
|---|---|---|
| **Age** | | |
| <45 | 178 | 17.84 |
| 45-65 | 441 | 44.19 |
| ≥66 | 379 | 37.98 |
| **Sex** | | |
| Male | 546 | 54.71 |
| Female | 452 | 45.29 |
| **Occupation** | | |
| Unemployed | 281 | 28.16 |
| Employed | 616 | 61.72 |
| Retired | 101 | 10.12 |
| **Educational status** | | |
| No education | 705 | 70.64 |
| Basic | 40 | 4.01 |
| SHS | 105 | 10.52 |
| Tertiary | 148 | 14.83 |
| **Ethnicity** | | |
| Dagombas | 521 | 52.20 |
| Gonjas | 35 | 3.51 |
| Others[1] | 442 | 44.29 |
| **Family history of stroke** | | |
| No | 966 | 96.79 |
| Yes | 32 | 3.21 |
| **Smoking** | | |
| No | 937 | 93.89 |
| Yes | 61 | 6.11 |
| **Alcohol intake** | | |
| No | 882 | 88.38 |
| Yes | 116 | 11.62 |
| **Temperature (°C)** | | |
| <37.5 | 859 | 86.07 |
| ≥37.5 | 139 | 13.93 |
| **Systolic BP (mmHg)** | | |
| <130 | 311 | 31.16 |
| ≥130 | 687 | 68.84 |
| **Diastolic BP (mmHg)** | | |
| <80 | 312 | 31.26 |
| ≥80 | 686 | 68.74 |
| **Heart rate (bpm)** | | |
| ≤100 | 371 | 37.17 |
| >100 | 627 | 62.83 |
| **Respiratory rate (breaths/min)** | | |
| ≤20 | 548 | 54.91 |
| >20 | 450 | 45.09 |

*(Continued)*

**Table 1.** (Continued)

| Characteristics | Frequency (N = 998) | Percentage (%) |
|---|---|---|
| **GCS level** | | |
| <9 | 274 | 72.55 |
| ≥9 | 724 | 27.45 |
| **Hypertension** | | |
| No | 214 | 21.44 |
| Yes | 784 | 78.56 |
| **Diabetes** | | |
| No | 787 | 78.86 |
| Yes | 211 | 21.14 |
| **Kidney disease** | | |
| No | 975 | 97.70 |
| Yes | 23 | 2.30 |
| **Pneumonia** | | |
| No | 635 | 63.63 |
| Yes | 363 | 36.37 |
| **Stroke type** | | |
| Ischemic | 516 | 51.70 |
| Haemorrhagic | 482 | 48.30 |
| **Atrial fibrillation** | | |
| No | 995 | 99.70 |
| Yes | 3 | 0.30 |
| **Right upper limb weakness** | | |
| No | 672 | 67.33 |
| Yes | 326 | 32.67 |
| **Left upper limb weakness** | | |
| No | 710 | 71.14 |
| Yes | 288 | 28.86 |
| **Right lower limb weakness** | | |
| No | 641 | 64.23 |
| Yes | 357 | 35.77 |
| **Left lower limb weakness** | | |
| No | 674 | 67.54 |
| Yes | 324 | 32.46 |
| **Seizures** | | |
| No | 869 | 87.07 |
| Yes | 129 | 12.93 |
| **Aphasia** | | |
| No | 653 | 65.43 |
| Yes | 345 | 34.57 |
| **Dysarthria** | | |
| No | 848 | 84.97 |
| Yes | 150 | 15.03 |
| **Dysphagia** | | |
| No | 849 | 85.07 |
| Yes | 149 | 14.93 |

*(Continued)*

**Table 1.** (Continued)

| Characteristics | Frequency (N = 998) | Percentage (%) |
|---|---|---|
| **Urea (mmol/L)** | | |
| Median (IQR) | 5.9 (4.82) | |
| **Creatinine(μmolL)** | | |
| Median (IQR) | 82.6 (60.30) | |
| **Bilirubin (μmolL)** | | |
| Median (IQR) | 11.5 (11.99) | |
| **Random blood sugar (mmol/L)** | | |
| Median (IQR) | 7.7(3.20) | |

1 = Wala, Kusasi, Frafra;

[1]basic (1–9 years of formal education, primary and junior high school), SHS (Senior High School, 10–12 years of formal education), and Tertiary (≥13 years, post-secondary education).

A structured checklist, developed in accordance with the RECORD (Reporting of studies Conducted using Observational Routinely collected Data) guidelines for studies using routinely collected health data, was used to extract information from LHIMS and the physical registers. The checklist ensured systematic and standardised retrieval of key variables, including sociodemographic characteristics, stroke type, comorbidities, and treatment outcomes for all patients admitted to the medical ward with a confirmed stroke diagnosis.

## Data analysis

Data extracted from LHIMS were entered into Excel, cleaned, and subsequently exported to STATA V.17 for analysis, an appropriate and robust platform for time-to-event analyses [20]. Univariable descriptive statistics, including frequencies and percentages, were presented in tables, with means and standard deviations for normally distributed continuous variables, and medians with interquartile ranges for skewed variables. All continuous variables were assessed for normality using the Shapiro–Wilk test, and their distributions were visualised using histograms. Kaplan–Meier survival curves were used to estimate time to in-hospital death, with discharge treated as a censoring event, and to describe overall survival probabilities. A log-rank test was performed to assess the difference between sex (male and female).

A stepwise Cox proportional hazards regression model was applied to identify variables associated with in-hospital mortality. Hazard ratios were used to measure the strength of these associations. The Schoenfeld residual test was conducted to verify the proportional hazards assumption, with variables meeting a p-value > 0.05 considered to satisfy the assumption. A bivariable Cox proportional hazards model was fitted for all predictors and variables with a p-value of less than 0.25 in order not to leave out important variables that may influence the outcome, and these were selected for inclusion in the stepwise multivariable Cox regression model. Multicollinearity among the predictors included in the multivariable model was assessed using the Variance Inflation Factor (VIF), and predictors with a VIF < 5 were included in the final model. [Mean VIF = 1.64, Max VIF = 2.63, Min VIF = 1.05], indicating no significant multicollinearity among the predictors. Statistical significance was set at p-value < 0.05 and a 95% confidence interval (CI).

## Patient and Public Involvement

Patients and members of the public were not involved in the design, conduct, reporting, or dissemination of this study, as it relied entirely on retrospectively collected hospital records.

## Results

### Baseline demographic and clinical characteristics of stroke patients

Most patients were aged between 45 and 65 years (44.19%). More than half of the patients were male (54.71%). Regarding occupation, most were employed (61.72%). Majority had no formal education (70.64%). Ethnically, Dagombas represented the largest group (52.20%). Most patients reported no family history of stroke (96.79%), were non-smokers (93.89%), and did not consume alcohol (88.38%). Clinically, most patients presented with a temperature below 37.5°C (86.07%) and had systolic and diastolic blood pressures of ≥130 mmHg (68.84%) and ≥80 mmHg (68.74%), respectively. A heart rate above 100 bpm was observed in 62.83% of patients, while 54.91% had a respiratory rate of 20 or fewer breaths per minute. The majority of patients had a Glasgow Coma Scale (GCS) score of ≥9 (72.55%). Hypertension was the most common, affecting 78.56% of patients, followed by diabetes (21.14%). Only a small proportion had kidney disease (2.30%). Pneumonia was documented in 36.37% of cases. Ischemic stroke was slightly more common than haemorrhagic stroke, accounting for 51.70% and 48.30% of cases, respectively. Atrial fibrillation was rare, reported in only 0.30% of patients. Regarding motor function and neurological deficits, the most common presentation was right upper limb weakness (32.67%), followed by left upper limb weakness (28.86%). Right lower limb weakness was observed in 35.77% of patients, while left lower limb weakness occurred in 32.46%. Seizures were noted in 12.93% of patients. Aphasia was present in 34.57% of cases, dysarthria in 15.03%, and dysphagia in 14.93% (**Table 1**). Approximately 60.62% of patients survived their admission, while 39.38% died during hospitalisation (**Fig 2**).

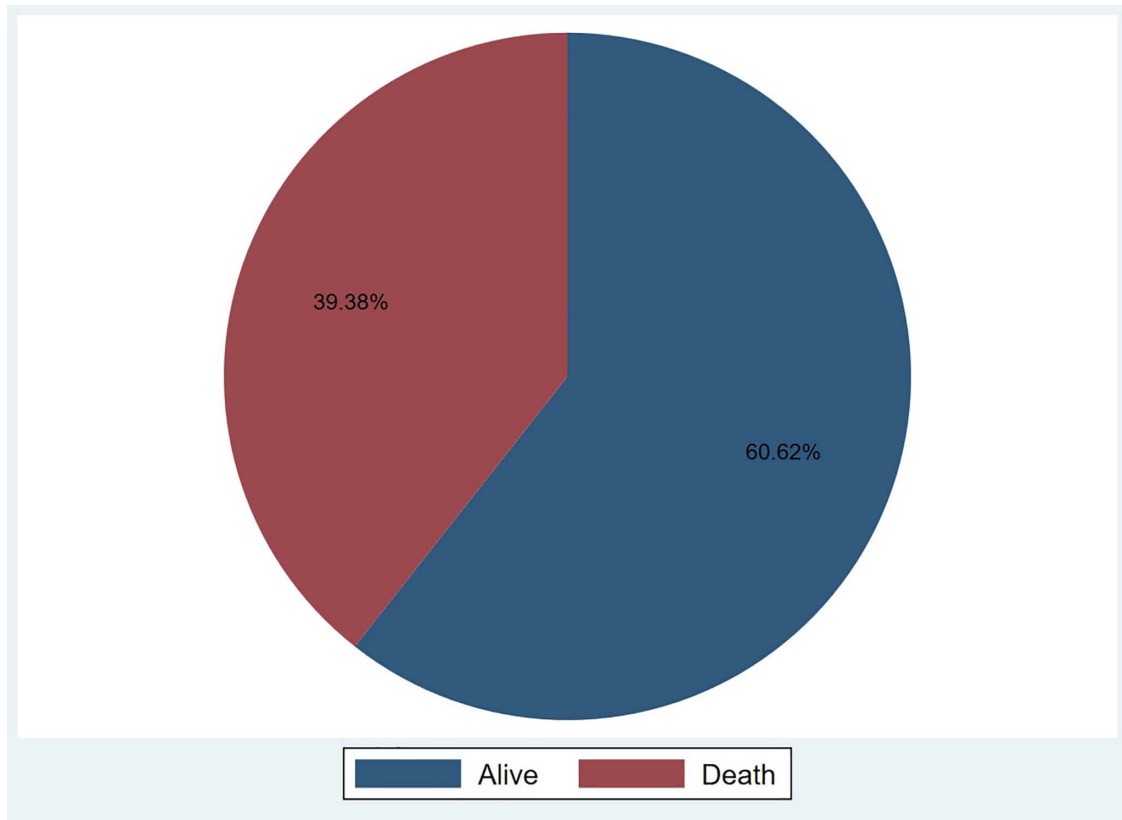

**Fig 2. Admission outcome of stroke patients at TTH.**

## Survival probability of stroke patients

The overall survival probability was 21.0%, with a mean survival time of 67 days (95% CI: 42–91) (**Fig 3**). The results also show a decline in survival rates with longer hospital stays. The probability of survival was 90.0% on the 1st day, 71.0% on the 5th day, and 60.0% on the 10th day. By the end of the 50-day observation period, although some patients continued to survive, the survival probability had dropped significantly to 42.5%.

Hemorrhagic stroke patients show shorter survival times than ischemic stroke patients, with a median survival of 31 days for ischemic stroke and 14 days for hemorrhagic stroke (**Fig 4**).

Throughout the follow-up period, males consistently demonstrate a higher survival probability than females. The survival rate for females declined sharply, indicating a higher mortality rate over the same timeframe (**Fig 5**). For instance, on the 5th day, the survival probability for females was 68.0%, compared to 73.0% for males, and on the 10th day, it was

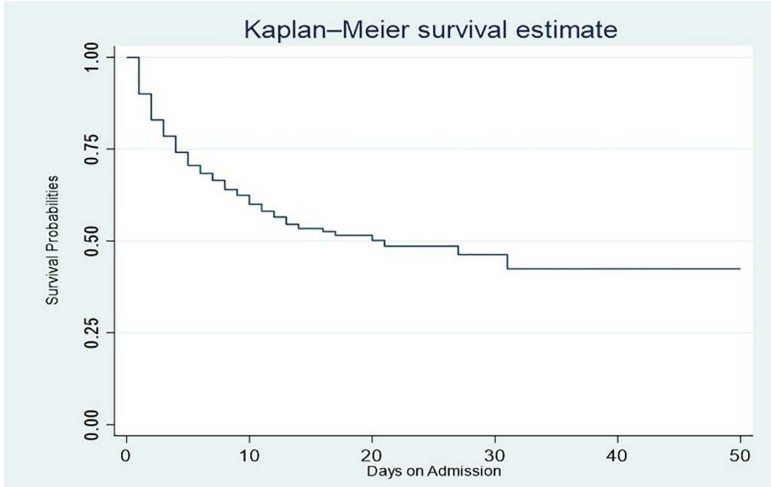

**Fig 3. Overall Kaplan-Meier survival curve of stroke patients at TTH.**

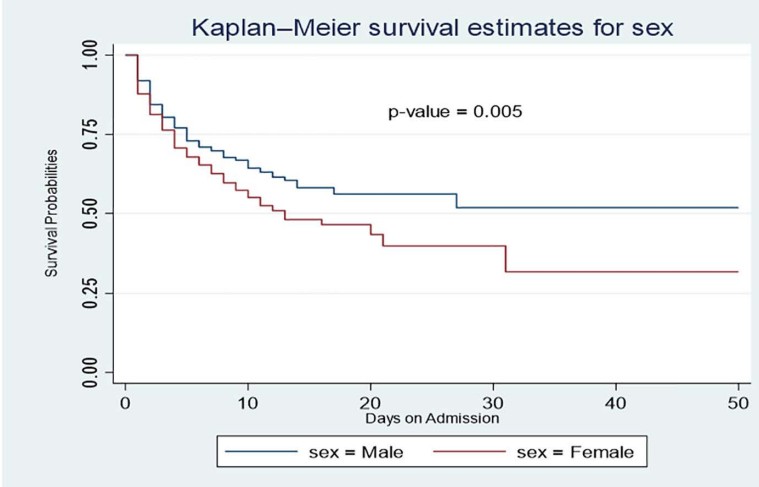

**Fig 4. Kaplan-Meier survival analysis by stroke type.**

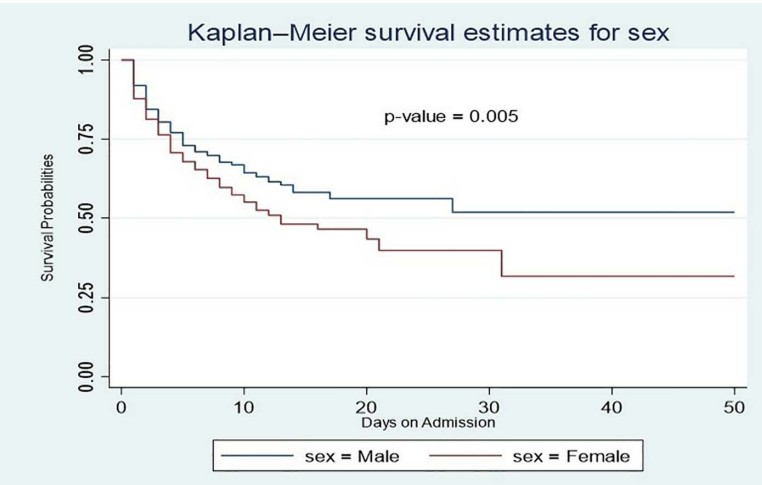

**Fig 5. Kaplan-Meier survival analysis by sex.**

64.0% for males and 55.0% whereas for females. The median survival time for females was 13 days (95% CI: 10–31), while for males, it was 17 days.

## Sociodemographic and clinical predictors of in-hospital mortality among stroke patients

Patients aged 45–65 years had a significantly higher hazard of death compared to those younger than 45 years (AHR = 1.28; 95% CI: 1.03–1.58). Female patients had a significantly 33% higher risk of mortality compared to males (AHR = 1.33; 95% CI: 1.07–1.65). Educational status was also significantly associated with mortality. Patients with basic education had a 54% lower risk of death compared to those with no education. Among ethnic groups, Dagombas had a 31% increased risk of death compared to other groups (AHR = 1.31; 95% CI: 1.05–1.62). Patients with diabetes had a 39% lower risk of mortality compared to non-diabetics (AHR = 0.62; 95% CI: 0.47–0.82). Similarly, individuals presenting with right lower limb weakness had a 33% lower risk of death than those without the condition (AHR = 0.67; 95% CI: 0.53–0.85). Patients with elevated temperature (≥ 37.5°C) had a 79% higher hazard of death (AHR = 1.79; 95% CI: 1.37–2.33), while those with pneumonia had a 1.59 times higher risk of mortality than patients without pneumonia (AHR = 1.59; 95% CI: 1.27–1.97). Additionally, patients with systolic blood pressure ≥ 130 mmHg exhibited a 32% higher risk of death compared to those with lower systolic pressure (AHR = 1.35, 95% CI: 1.06–1.73). Interestingly, a heart rate above 100 bpm was associated with a significantly lower risk of mortality (AHR = 0.50, 95% CI: 0.40–0.62), as was a respiratory rate above 20 (AHR = 0.63, 95% CI: 0.49–0.79). These associations may reflect differences in stroke subtype or patient management and should be interpreted with caution. Higher oxygen saturation was significantly associated with better survival (AHR = 0.96, 95% CI: 0.95–0.98), while for each mmol/L increase in urea, the risk of mortality increased by 2% (AHR = 1.02; 95% CI: 1.01–1.03 per mmol/L) (Table 2).

## Discussion

This study investigated the in-hospital survival rate and predictors of mortality among stroke patients admitted to the Tamale Teaching Hospital, Ghana. The study found that several factors, including female sex, Dagomba ethnicity, elevated temperature, pneumonia, higher systolic blood pressure, and increased urea levels, were associated with a higher risk of death due to stroke. Conversely, basic education, diabetes, right lower limb weakness, elevated heart and respiratory rates, and higher oxygen saturation were linked to improved survival outcomes.

**Table 2. Sociodemographic and clinical predictors of in-hospital mortality among stroke patients.**

| Characteristics | Crude HR | 95% CI | p-value | Adjusted HR | 95% CI | p-value |
|---|---|---|---|---|---|---|
| **Age group** | | | | | | |
| <45 | Ref | | | Ref | | |
| 45-65 | 1.25 | 0.92-1.70 | 0.154 | 1.28 | 1.03-1.58 | 0.027 |
| ≥66 | 1.07 | 0.78-1.48 | 0.663 | | | |
| **Sex** | | | | | | |
| Male | Ref | | | Ref | | |
| Female | 1.34 | 1.08-1.66 | 0.008 | 1.33 | 1.07-1.65 | 0.010 |
| **Educational status** | | | | | | |
| No education | Ref | | | Ref | | |
| Basic | 0.49 | 0.24-0.98 | 0.044 | 0.46 | 0.23-0.93 | 0.039 |
| SHS | 0.95 | 0.67-1.36 | 0.669 | | | |
| Tertiary | 0.57 | 0.40-0.83 | 0.003 | | | |
| **Ethnicity** | | | | | | |
| Others[1] | Ref | | | Ref | | |
| Gonjas | 0.62 | 0.29-1.33 | 0.219 | | | |
| Dagombas | 1.34 | 1.08-1.67 | 0.008 | 1.31 | 1.05-1.62 | 0.018 |
| **Alcohol intake** | | | | | | |
| No | Ref | | | Ref | | |
| Yes | 0.75 | 0.52-1.07 | 0.113 | 0.69 | 0.48-0.99 | 0.044 |
| **Temperature (°C)** | | | | | | |
| <37.5 | Ref | | | Ref | | |
| ≥37.5 | 2.26 | 1.75-2.91 | <0.001 | 1.79 | 1.37-2.33 | <0.001 |
| **Systolic BP (mmHg)** | | | | | | |
| <130 | Ref | | | Ref | | |
| ≥130 | 1.45 | 1.14-1.85 | 0.003 | 1.35 | 1.06-1.73 | 0.017 |
| **Diastolic BP (mmHg)** | | | | | | |
| <80 | Ref | | | | | |
| ≥80 | 1.45 | 1.13 −1.85 | 0.003 | | | |
| **Heart rate (bpm)** | | | | | | |
| ≤100 | Ref | | | Ref | | |
| >100 | 0.40 | 0.32-0.49 | <0.001 | 0.52 | 0.42-0.66 | <0.001 |
| **Respiratory rate (breaths/min)** | | | | | | |
| <20 | Ref | | | Ref | | |
| ≥20 | 0.51 | 0.41-0.64 | <0.001 | 0.60 | 0.47-0.76 | <0.001 |
| **GCS level** | | | | | | |
| ≥9 | Ref | | | | | |
| <9 | 1.23 | 0.97-1.53 | 0.094 | | | |
| **Diabetes** | | | | | | |
| No | Ref | | | Ref | | |
| Yes | 0.73 | 0.55-0.95 | 0.021 | 0.62 | 0.47-0.82 | 0.001 |
| **Pneumonia** | | | | | | |
| No | Ref | | | Ref | | |
| Yes | 1.89 | 1.53-2.34 | <0.001 | 1.59 | 1.27-1.97 | <0.001 |
| **Stroke type** | | | | | | |
| Ischaemic | Ref | | | | | |
| Haemorrhagic | 1.17 | 0.95-1.45 | 0.144 | | | |

*(Continued)*

**Table 2.** (Continued)

| Characteristics | Crude HR | 95% CI | p-value | Adjusted HR | 95% CI | p-value |
|---|---|---|---|---|---|---|
| **Right upper limb weakness** | | | | | | |
| No | Ref | | | | | |
| Yes | 0.76 | 0.60-0.96 | 0.025 | | | |
| **Right lower limb weakness** | | | | | | |
| No | Ref | | | | | |
| Yes | 0.72 | 0.57-0.91 | 0.005 | 0.67 | 0.53-0.85 | 0.001 |
| **Dysarthria** | | | | | | |
| No | Ref | | | | | |
| Yes | 0.69 | 0.49-0.96 | 0.030 | | | |
| **Urea (mmol/L)** | | | | | | |
| | 1.02 | 1.01-1.03 | <0.001 | 1.02 | 1.01-1.03 | <0.001 |
| **Creatinine (µmolL)** | | | | | | |
| | 1.00 | 1.00-1.00 | <0.001 | | | |
| **Bilirubin (µmolL)** | | | | | | |
| | 1.01 | 0.99-1.02 | 0.196 | | | |
| **Random blood sugar (mmol/L)** | | | | | | |
| | 1.01 | 1.00-1.02 | 0.022 | | | |

**HR = Hazard ratio, Ref = reference group, CI =Confidence interval, 1 = Wala, Kusasi, Frafra**

The overall survival probability was high in the initial days of admission but decreased as follow-up time increased. Revealing further that a high number of patients died in the first 10 days after being diagnosed with stroke. This is comparable to a study in North West Ethiopia, with similar observations [4]. Also, the median survival time of 21 days was lower than in a study in Ethiopia that had a median survival time of 41 days [21]. This difference is likely explained by contextual factors such as variation in stroke severity and complication rates, health systems, and variation in sample size [4,21].

Males consistently showed a higher survival probability throughout the follow-up period, indicating a gender disparity in stroke outcomes. This aligns with findings from a study by [22] conducted in South Korea. The steep decline in survival rates among females, indicates a higher mortality rate, evident as early as the 5th-day post-stroke, where the survival probability for females was 68.0% compared to 73.0% for males. This pattern persists, with a significant gap by the 10th day, as survival probabilities drop to 55.0% for females and 64.0% for males. The median survival time further underscores this difference, with females showing a notably shorter median survival time of 13 days compared to 17 days for males. This disparity may be attributed to more severe post-stroke complications or other factors impacting female survival rates over time [23].

The findings also revealed that the odds of mortality were high among females compared to males. This aligns with a study in Ghana at the Korlebu Teaching Hospital [24]. Several factors may contribute to the higher burden of stroke-related mortality and disability among females. Sociocultural gender roles and biological differences influence stroke risk, assessment, treatment, and outcomes. The relationship between general stroke risk factors and female-specific risk variables differs considerably. Additionally, there are differences in how women experience stroke symptoms, respond to treatment, and recover after a stroke compared to men [23].

The study also revealed that stroke patients with no formal education faced a higher risk of mortality compared to those with education. Specifically, having a basic education was associated with approximately a 54% lower risk of death. This finding is consistent with studies conducted in Ghana and China [14,25]. This finding may be attributed to individuals

with no education often displaying less healthy lifestyle behaviours and greater clinical risk factors for stroke, a pattern observed in both genders [26].

Also, the study found that individuals of Dagomba ethnicity had a higher risk of mortality following stroke. While direct comparisons are limited, this observation aligns with findings from the United States, where studies have consistently reported higher stroke mortality rates among Black populations [27,28]. A possible explanation is that cultural beliefs within certain ethnic groups may encourage reliance on traditional healing practices, leading to delays in seeking hospital care [29,30]. These disparities include a lack of awareness about stroke symptoms, delays in seeking timely treatment, and limited understanding of risk factors. Differences in attitudes, beliefs, and adherence to medical advice also vary across races and ethnicities.

The study revealed that pneumonia was also a predictor of mortality. The risk of mortality was high among stroke patients with pneumonia complications. This is consistent with a study in Ghana, Nigeria and Ethiopia [14,21,31]. The increased risk of mortality can be linked to several factors. Post-stroke patients frequently encounter swallowing difficulties and limited mobility, both of which can contribute to the development of lung infections like pneumonia [32]. Moreover, strokes can induce an immune response that heightens vulnerability to infections, disrupts the tracheal epithelium, diminishes lung clearance, and hampers the expulsion of secretions, further elevating the likelihood of pneumonia [33].

Interestingly, the study revealed that stroke patients with diabetes had a lower risk of mortality, which contrasts with findings observed in Ghana [14,34]. This finding needs to be further investigated as it is very counterintuitive unless something special is being done in the population or the health facility within which these cases are being picked

Furthermore, we found that patients presenting with right lower limb weakness had a lower risk of death compared to those without such weakness. Although this specific finding has limited direct support in existing literature, a study conducted in Canada reported that outcomes tend to be more favourable in individuals with motor strokes, particularly those presenting with monoparesis, defined as weakness in a single limb [35]. While this does not directly align with our findings, it suggests that isolated motor deficits may be associated with less severe stroke presentations, potentially contributing to improved survival. Overall, patients with motor stroke symptoms often experience fewer complications and shorter hospital stays, which may partly explain the observed association in our study. Also, these patients are more likely to experience symptom resolution by the time of hospital discharge, which further supports the observation that isolated motor deficits may predict better prognoses compared to more extensive motor involvement.

### Implications for practice and research

Despite the limitations, the findings of this study provide an important contribution with relevant implications for improving the survival of stroke patients. The identification of both demographic and clinical predictors of stroke mortality has important implications for clinical practice. High-risk patients, such as females, individuals with low or no education, those presenting with aphasia, pneumonia, elevated temperature, or high systolic blood pressure, may benefit from closer monitoring and aggressive management during the acute phase of stroke. The protective associations observed with diabetes suggest potential responsiveness to treatment that warrants further investigation. Early identification and prompt intervention for modifiable risk factors like diabetes could improve survival outcomes, especially during the critical first 10 days post-admission when mortality risk is highest.

### Strengths and limitations

This study provides valuable insights into the factors that influence survival among stroke patients in a real-world clinical setting. By using a multivariable Cox regression model, we were able to control for several demographic and clinical variables, helping us identify the key predictors of mortality more accurately. The study also benefits from a relatively large sample size and a three-year follow-up period, which strengthens the reliability of our findings. Importantly, by examining both social and clinical factors, we offer a more holistic picture of the challenges stroke patients face in our context.

That said, our study has a few limitations worth noting. Because it is based on retrospective data, we were limited to the information recorded in patients' medical files, which were sometimes incomplete or missing key details. We also could not assess stroke severity scores (e.g., NIHSS, mRS) or functional outcomes at admission, which might have further clarified the survival patterns we observed. Although Kaplan–Meier survival analysis was used to compare survival between ischaemic and haemorrhagic stroke patients, this study did not conduct subtype-specific multivariable regression analyses to identify predictors of mortality within each stroke subtype. Also, detailed information on arterial territories in ischaemic stroke and cerebral localization of haemorrhagic stroke was not available. These limitations may have reduced the ability to detect subtype-specific prognostic factors. In addition, we only looked at what happened during the hospital stay, so we might have missed deaths that occurred after discharge.

## Conclusion

The study revealed that stroke patients treated at the Tamale Teaching Hospital face a declining probability of survival over time, with the highest risk of death occurring within the first 10 days of admission. To reduce preventable deaths, health system actions are needed including: (1) establishing or strengthening dedicated stroke care pathways or a stroke unit at TTH to ensure rapid assessment and standardized management; (2) prioritizing early monitoring and control of modifiable, high-risk comorbidities such as fever, pneumonia, and elevated systolic blood pressure; (3) implementing targeted screening and public-health outreach for high-risk groups and tailored education for patients and caregivers; and (4) ensuring protocols for infection prevention and early rehabilitation and discharge planning to reduce complications and length of stay. However, given the retrospective design and potential for residual confounding, large and prospective studies are still needed.

## Supporting information

**S1 File. Data used for the analysis.**
(XLSX)

## Acknowledgments

We acknowledge the Senior Health Information Officer at TTH, Mr. Abdul–Hafiz Zakari, for his generous assistance and cooperation during the research.

## Author contributions

**Conceptualization:** Sulemana Baba Abdulai, Yeetey Enuameh.

**Data curation:** Sulemana Baba Abdulai.

**Formal analysis:** Sulemana Baba Abdulai, Julius Kwabena Karikari.

**Methodology:** Sulemana Baba Abdulai, Yeetey Enuameh.

**Supervision:** Yeetey Enuameh.

**Writing – original draft:** Sulemana Baba Abdulai.

**Writing – review & editing:** Sulemana Baba Abdulai, Julius Kwabena Karikari, Penias Tembo, Theogene Habumugisha, William Tembo, Alhaji Ibrahim Cobbinah, Yeetey Enuameh.

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
