## [Decision Letter · Decision Letter 0]

4 Sep 2025

PONE-D-25-39162

PREDICTORS OF SURVIVAL OF STROKE PATIENTS IN THE TAMALE TEACHING HOSPITAL IN GHANA: A 3-YEAR RETROSPECTIVE REVIEW

PLOS ONE

Dear Dr. Abdulai,

Thank you for submitting your manuscript to PLOS ONE. After careful consideration, we feel that it has merit but does not fully meet PLOS ONE’s publication criteria as it currently stands. Therefore, we invite you to submit a revised version of the manuscript that addresses the points raised during the review process.

We look forward to receiving your revised manuscript.

Kind regards,

Mehdi Sharafi, assistant professor

Academic Editor

PLOS ONE

Journal Requirements:

2. In the online submission form, you indicated that [The anonymised datasets used and/or analysed during the current study are not publicly accessible; however, they can be obtained from the corresponding author upon reasonable request.].

Reviewers' comments:

Reviewer's Responses to Questions

**Comments to the Author**

1. Is the manuscript technically sound, and do the data support the conclusions?

Reviewer #1: Yes

Reviewer #2: Yes

2. Has the statistical analysis been performed appropriately and rigorously? 

Reviewer #1: Yes

Reviewer #2: Yes

3. Have the authors made all data underlying the findings in their manuscript fully available?

Reviewer #1: Yes

Reviewer #2: Yes

4. Is the manuscript presented in an intelligible fashion and written in standard English?

Reviewer #1: Yes

Reviewer #2: Yes

5. Review Comments to the Author

Reviewer #1: This study is very standard and has appropriate structure and content. However, there are a few minor methodological flaws in the introduction, methods, and discussion that have been commented on in the article.

Reviewer #2: Overall Evaluation

This study investigates predictors of survival among stroke patients admitted to Tamale Teaching Hospital (Ghana). The topic is highly relevant, as survival data on stroke patients in sub-Saharan Africa are scarce. The manuscript is valuable and based on a relatively large dataset, but several methodological clarifications, improvements in tables, and more cautious interpretation of results are needed before publication.

Abstract

- The follow-up period (mean survival: 67 days) is not mentioned.

- A protective effect of diabetes is reported as an unexpected finding, but it is presented without sufficient explanation or cautious interpretation.

- Suggestion: The follow-up duration and key study limitations (e.g., retrospective design) should be stated. Unexpected findings should be reported with greater caution and accompanied by interpretation.

Methods

- The definition of the outcome is unclear. The text indicates that “survival status” was determined at discharge (alive or dead), but the Kaplan–Meier results report a mean survival of 67 days with follow-up up to 50 days. This creates confusion as to whether the authors considered in-hospital survival only or included post-discharge follow-up.

- Suggestion: Authors should explicitly clarify how the outcome was defined (discharge status only or extended follow-up). Without this, it is difficult to compare results with other studies and to interpret the Kaplan–Meier curves.

- Handling of missing data is not described.

- No justification for sample size or study power is provided.

- Suggestion:

• Clarify the approach to handling missing data.

• Acknowledge limitations due to the absence of stroke severity measures (e.g., NIHSS, mRS).

Results and Tables

- The results are generally clear, but several issues require attention:

• Some associations (e.g., protective effect of diabetes, HR >100 bpm) need further clarification.

• The Kaplan–Meier survival curve should be more fully described (e.g., stratified by stroke type).

• Table 1: For the variable educational level, it should be clarified how many years of formal education each category represents (e.g., Basic, SHS, Tertiary). Abbreviations (such as SHS) should also be explained in the table footnotes or text for the benefit of international readers.

Conclusion

- The current conclusion largely repeats the results.

- Suggestion: The conclusion should instead emphasize policy and clinical implications, such as the need for stroke units, closer monitoring of high-risk patients, and targeted screening for risk factors.

References

- The reference list combines local and international sources but includes some outdated ones. In particular, Hatano 1976, which refers to the old WHO definition of stroke, is outdated.

- Suggestion: This reference should be replaced or supplemented with more recent and comprehensive sources, such as the AHA/ASA 2013 statement or more recent systematic reviews on stroke definition.

6. PLOS authors have the option to publish the peer review history of their article (what does this mean?). If published, this will include your full peer review and any attached files.

Reviewer #1: No

Reviewer #2: **Yes: **Hassan Karami

---

## [Author Response · Author response to Decision Letter 1]

6 Oct 2025

Reviewers Comments Author Responses

Reviewer #1

The title needs to be rewritten We revised the title to clarify the outcome and study design. Suggested title: “In-hospital survival and predictors of mortality among stroke patients at a Tertiary Hospital in Ghana: a 3-year retrospective cohort study.”

Summarise the abstract introduction We appreciate the reviewer’s suggestion to make the abstract more concise. In response, we have summarised the Introduction section of the abstract to focus on the essential background and study objective. The revised text now reads:

“Stroke is a leading cause of death and disability globally and imposes a growing burden on Ghana’s healthcare system. This study aimed to estimate in-hospital survival rates and identify predictors of mortality among stroke patients admitted to a tertiary hospital in Ghana over a 3-year period.”

Your literature review is weak. Please cite examples of similar studies. We appreciate the reviewer’s feedback. In response, we have strengthened the literature review by including additional studies on stroke survival and its determinants. Specifically, we have now cited:

• Sarfo FS, et al. Long-Term Outcomes of Stroke in a Ghanaian Outpatient Clinic. J Stroke Cerebrovasc Dis. 2018.

• Sarfo FS, et al. Key determinants of long-term post-stroke mortality in Ghana.(line 91)

Provide logical reasoning for all comparisons (e.g., why your median survival differs from other studies). We rewrote the comparison paragraph to state explicitly why median survival may differ across studies: This difference is likely explained by contextual factors such as variation in stroke severity and complication rates, health systems, and variation in sample size (lines 271-273)

Reviewer #2

1. The follow-up period (mean survival: 67 days) is not mentioned. Thank you for pointing this out. We have revised the Methods section of the abstract to include the mean follow-up duration (Line 35). It now reads: “Kaplan–Meier survival analysis was used to estimate the survival rate of stroke patients (mean follow-up: 67 days).”

2. A protective effect of diabetes is reported as an unexpected finding, but it is presented without sufficient explanation or cautious interpretation. We appreciate this important comment. In the revised abstract, we have reported the association between diabetes mellitus and mortality more cautiously, noting that the apparent lower risk may be due to differences in case management or residual confounding (lines 50 – 53). The Results section now states: “Diabetes mellitus showed an apparently lower risk of mortality (AHR = 0.62, 95% CI: 0.47–0.82); this finding should be interpreted cautiously as it may reflect differences in case management or residual confounding.” (lines 252-254)

The definition of the outcome is unclear. The text indicates that “survival status” was determined at discharge (alive or dead), but the Kaplan–Meier results report a mean survival of 67 days with follow-up up to 50 days. This creates confusion as to whether the authors considered in-hospital survival only or included post-discharge follow-up. Suggestion: Authors should explicitly clarify how the outcome was defined (discharge status only or extended follow-up). Without this, it is difficult to compare results with other studies and to interpret the Kaplan–Meier curves. Thank you for this important observation. We have revised the Materials and Methods section to explicitly state that survival status was determined at hospital discharge only; no post-discharge follow-up was included. The time-to-event variable used for Kaplan–Meier analysis was calculated as the number of days from admission to discharge or in-hospital death. This clarification has been added to the “Study Variables” (lines 129-132) and “Data Analysis” (lines 151-152) subsections.

3. The follow-up duration and key study limitations (e.g., retrospective design) should be stated. Unexpected findings should be reported with greater caution and accompanied by interpretation. We have incorporated the mean follow-up time in the Methods section, added a sentence on the retrospective design as a limitation, and rephrased the Conclusion to reflect the cautious interpretation of unexpected findings. The Conclusion now reads: “Given the retrospective design and possible unmeasured confounders, the observed lower mortality risk among diabetic patients should be interpreted with caution, and prospective studies are warranted to confirm these associations.”

Handling of missing data is not described. We appreciate this comment. We have now included a statement describing how missing data were handled. Specifically, patients with incomplete or missing key variables were excluded from the analysis, and the number of excluded cases is reported in Figure 1

No justification for sample size or study power is provided.

We thank the reviewer for this observation. We have added a sentence acknowledging that no a priori sample size or power calculation was performed because the study included all eligible stroke cases within the 3-year period (census sampling). This has been added to the “Patient Identification” subsection (lines 113-114).

Suggestion: Acknowledge limitations due to the absence of stroke severity measures (e.g., NIHSS, mRS). We agree with this suggestion. This has been highlighted in the discussion of limitations.

Some associations (e.g., protective effect of diabetes, HR >100 bpm) need further clarification. We thank the reviewer for this observation. We have added text in the Results section noting that the apparent protective effects of diabetes, elevated heart rate (>100 bpm), and higher respiratory rate may reflect differences in case management, stroke subtype, or residual confounding (lines 250-252). We caution readers that these findings should be interpreted carefully and may not indicate true protective effects.

The Kaplan–Meier survival curve should be more fully described (e.g., stratified by stroke type). We appreciate this suggestion. We have added a description of the Kaplan–Meier curves stratified by stroke type (ischemic vs. hemorrhagic) in the Results section. A new figure (Figure 4) has been included showing separate curves for each type with log-rank p-values reported.

Table 1: For the variable educational level, it should be clarified how many years of formal education each category represents (e.g., Basic, SHS, Tertiary). Abbreviations (such as SHS) should also be explained in the table footnotes or text for the benefit of international readers. Thank you for this important point. We have now clarified in Table 1 footnotes what each educational category represents: Basic (1–9 years of formal education, primary and junior high school), SHS (10–12 years of formal education, senior high school), and Tertiary (≥13 years, post-secondary education). All abbreviations including SHS have been explained in the footnotes.

The current conclusion largely repeats the results.

Suggestion: The conclusion should instead emphasize policy and clinical implications, such as the need for stroke units, closer monitoring of high-risk patients, and targeted screening for risk factors. Thank you. We have rewritten the Conclusion to focus on policy and clinical implications rather than restating results. The revised Conclusion emphasizes the need to strengthen acute stroke care at TTH (including establishing a dedicated stroke unit or protocolized acute stroke pathway), enhanced monitoring and early management of high-risk features (fever, pneumonia, elevated systolic BP), targeted screening and community education, and further prospective research to confirm observed associations and investigate ethnic and sex-based disparities (lines 365-374).

The reference list combines local and international sources but includes some outdated ones. In particular, Hatano 1976, which refers to the old WHO definition of stroke, is outdated.

Suggestion: This reference should be replaced or supplemented with more recent and comprehensive sources, such as the AHA/ASA 2013 statement or more recent systematic reviews on stroke definition. We agree and have replaced the Hatano (1976) citation with the AHA/ASA 2013 expert consensus statement, 'An Updated Definition of Stroke for the 21st Century'. (line 60)

---

## [Decision Letter · Decision Letter 1]

24 Nov 2025

PONE-D-25-39162R1

IN-HOSPITAL SURVIVAL AND PREDICTORS OF MORTALITY AMONG STROKE PATIENTS AT A TERTIARY HOSPITAL IN GHANA: A 3-YEAR RETROSPECTIVE COHORT STUDY

PLOS ONE

Dear Dr. Abdulai,

Thank you for submitting your manuscript to PLOS ONE. After careful consideration, we feel that it has merit but does not fully meet PLOS ONE’s publication criteria as it currently stands. Therefore, we invite you to submit a revised version of the manuscript that addresses the points raised during the review process.p

We look forward to receiving your revised manuscript.

Kind regards,

Mickael Essouma, M. D.

Academic Editor

PLOS ONE

Journal Requirements:

Additional Editor Comments:

I have added some comments to this decision letter to further improve the manuscript.

Reviewers' comments:

Reviewer's Responses to Questions

**Comments to the Author**

1. If the authors have adequately addressed your comments raised in a previous round of review and you feel that this manuscript is now acceptable for publication, you may indicate that here to bypass the “Comments to the Author” section, enter your conflict of interest statement in the “Confidential to Editor” section, and submit your "Accept" recommendation.

Reviewer #1: (No Response)

Reviewer #2: All comments have been addressed

2. Is the manuscript technically sound, and do the data support the conclusions?

Reviewer #1: (No Response)

Reviewer #2: Yes

3. Has the statistical analysis been performed appropriately and rigorously? 

Reviewer #1: (No Response)

Reviewer #2: Yes

4. Have the authors made all data underlying the findings in their manuscript fully available?

Reviewer #1: (No Response)

Reviewer #2: Yes

5. Is the manuscript presented in an intelligible fashion and written in standard English?

Reviewer #1: (No Response)

Reviewer #2: Yes

6. Review Comments to the Author

Reviewer #1: The conclusion section of the abstract needs to be rewritten. It should be written more concisely and based on the findings, rather than repeating the findings.

The introduction section provides a poor review of the literature and does not clearly explain and present the research gap.

The type of study should be stated more clearly and precisely.

Reviewer #2: (No Response)

7. PLOS authors have the option to publish the peer review history of their article (what does this mean?). If published, this will include your full peer review and any attached files.

Reviewer #1: No

Reviewer #2: **Yes: **Dr. Hassan Karami

---

## [Author Response · Author response to Decision Letter 2]

12 Dec 2025

Reviewers Comments Author Responses

I suggest removing “3-year” in the manuscript’s title and perhaps replacing it with “50-day” because it gives the impression that the duration of follow-up was 3 years; meanwhile, this was likely a short-term cohort (50 days of follow-up) with patients being followed up only from the day of hospital admission for stroke to the day of discharge. The 3-year period is rather the study period

Thank you for the suggestion regarding the use of “3-year” in the title. we agree that including a time period may unintentionally imply a 3-year patient follow-up, which does not reflect the study design. Although the study used three years of retrospective hospital records, individual patients were followed only until discharge or death.

Additionally, the follow-up window was not fixed at 50 days; rather, it varied for each patient and ended at discharge or death. The mean survival time was 67 days, and so specifying a “50-day follow-up” in the title would inaccurately describe the study results.

To avoid misinterpretation while ensuring accuracy, we will remove “A 3-Year” from the title entirely.

Abstract. It includes 332 words whereas the max word count of abstracts in PLOS One abstracts articles is 300. Therefore, consider reducing the word count in the abstract section by decreasing for example, the length of the introduction sub-section of the abstract. Review the interpret ation of results of survival predictors

Thank you for the feedback regarding the length of the abstract. We appreciate the reminder about the 300-word limit for PLOS One abstracts. We have revised the abstract and reduced it from 332 words to 300 words, primarily by shortening the introductory section and tightening the presentation of the results. We also reviewed and refined the interpretation of the survival predictors to ensure clarity and accuracy.

I find it incredibly long given that stroke is relatively well known in medicine, even in Africa. Could you then reduce its length to a max of 1.5 pages? This would entail focusing on current knowledge on stroke in sub-Saharan Africa and Ghana, gaps in knowledge on stroke in sub-Saharan Africa and Ghana, and finally, the rationale for conducting this study, the study’s aim, and the overarching goal of the study. Thank you very much for this insightful comment. We appreciate the concern regarding the length of the Introduction section. While we agree that stroke is a well-known condition globally, the burden, patterns, and predictors of survival in sub-Saharan Africa, particularly in Ghana, remain significantly under-documented compared with other regions. For this reason, the Introduction aimed to provide sufficient contextual background on:

1. The epidemiology of stroke in sub-Saharan Africa,

2. Existing knowledge gaps regarding stroke outcomes and survival in Ghana, and

3. The justification for conducting a predictive survival analysis within this setting.

These elements were included to help readers, especially those unfamiliar with the Ghanaian or African context, understand the unique clinical and health-system factors that make stroke survival research in this region essential. However, we appreciate the

Editor’s suggestion on conciseness.

Did you use the RECORD guidelines for reporting a retrospective cohort (see the manuscript’s title) or a retrospective chart review (link:http://doi.org/10.3352/jeehp.2013.10.12) study? If the study was a retrospective

cohort, was it a predictive cohort as expected (https://doi.org/10.1186/s12982-018-0080-2)? Consider specifying.

Thank you for raising this important point regarding the reporting framework and study classification. The present study was conducted as a retrospective cohort study, where all patients admitted with a confirmed diagnosis of stroke between 2021 and 2023 were identified and followed until the outcome of discharge or death. Although the data were extracted from existing electronic records (LHIMS), the study design aligns more closely with a retrospective cohort than a chart review because:

1. A defined cohort was established based on clear eligibility criteria (all stroke admissions within the specified period).

2. Patients were followed over time from admission to discharge/death to determine survival outcomes.

3. Predictor variables were assessed in relation to a time-to-event outcome, which is consistent with a predictive survival cohort.

In response to the editor’s suggestion, we have clarified in the Methods section that the study followed the RECORD guidelines (Reporting of studies Conducted using Observational Routinely-collected Data), as recommended for retrospective cohort studies using routinely collected health information. The structured checklist used for data extraction was designed to ensure systematic and standardized retrieval of predictor and outcome variables from LHIMS and the patient registers.

This clarification has now been incorporated into the revised manuscript (lines 156-165).

Could you provide details about the sampling method used and sample size estimation (doi:10.1184/ratoil.22722015019)? Did you conduct the study in agreement with the Declaration of Helsinki?

Thank you for this comment. The study did not employ a sampling procedure because all eligible stroke cases admitted to the Tamale Teaching Hospital during the 3-year study period were included. This approach constitutes a census of all cases, not a sample; therefore, a formal sample size calculation was not applicable. The use of a census ensured complete coverage of all stroke admissions within the defined period, thereby minimizing selection bias and increasing the representativeness of the dataset.

Regarding ethical considerations, the study adhered to the principles of the Declaration of Helsinki for research involving human participants. Ethical approval was obtained from the appropriate institutional review board, and permission was granted by the hospital authorities to access patient records. Since the study involved retrospective review of existing records, no direct patient contact occurred, and all extracted data were anonymized prior to analysis to ensure confidentiality and privacy.

These clarifications have been added to the revised manuscript under the Methods section (lines 107-109, 113-114).

How did you handle missing data (10.1097/EDE.000000000000409)?

We appreciate this comment. We have now included a statement describing how missing data were handled. Specifically, patients with incomplete or missing key variables were excluded from the analysis, and the number of excluded cases is reported in Figure 1

Could you provide more details about the study setting (notably regarding their involvement in the management of stroke in Ghana: specialized unit or non-specialized facility, number of stroke cases managed, credentials of health professionals working there…)?

Thank you for this important comment. We have now expanded the description of the study setting to clarify the hospital’s role in stroke management in Ghana, including the type of facility, availability of specialized care, staffing, and the volume of stroke cases managed.

Tamale Teaching Hospital (TTH) is the main tertiary referral centre for northern Ghana and historically managed stroke patients in general medical wards. In September 2024, the hospital established its first dedicated stroke unit to improve coordinated stroke care. The hospital currently has clinicians with specialized training in neurology and stroke care, including a neurologist and a palliative care nurse specialist who support protocol development. As a high-volume referral facility, TTH manages substantial numbers of stroke patients; for example, 105 ischemic stroke admissions were recorded between January and October 2021 (lines 121-134)

How did you collect data on study predictors and outcomes? What was the duration of follow-up at which the outcome variable survival was recorded?

Thank you for this important comment. We agree that clarity regarding data collection and follow-up duration is essential. However, we would like to highlight that these details were already described in the Study Variables subsection (line 140).

Specifically, the manuscript explains that all predictors and outcomes were extracted from routinely collected patient data (LHIMS and physical patient registers) using a structured checklist. The outcome variable, in-hospital survival status, was defined at discharge as either “alive” or “dead.” As stated, the duration of follow-up was from the date of admission to the date of in-hospital death or discharge, which reflects the full length of each patient’s hospital stay. No post-discharge follow-up was available since the study was based on retrospective review of hospital records.

To further improve clarity in response to the reviewer’s suggestion, we have revised the Methods section to make these details more explicit and easier to locate, while keeping the core content unchanged

Data analysis sub-section: Consider providing a reference supporting appropriateness of STATA for the analyses performed

Thank you for this helpful suggestion. Stata is a widely used and well-validated statistical software for conducting survival analyses, including Kaplan–Meier estimation and Cox proportional hazards regression. In response to the reviewer’s comment, a supporting reference has now been added to the Data Analysis subsection.

For example, Stata has been described as an appropriate and robust platform for time-to-event analysis (e.g., Cleves et al., An Introduction to Survival Analysis Using Stata, Stata Press, 2010). This reference has been included (line 170)

Consider adding a sub-section titled “Patient and public involvement” at the end of this section. Is the first sub-section (see title in line 164) about baseline participant characteristics? Consider specifying.

Patient and Public Involvement:

Because this study was a retrospective review of existing hospital records, patients and the public were not involved in the design, conduct, reporting, or dissemination plans of the research. This is consistent with standard practice for retrospective cohort studies using routinely collected clinical data. In response to the reviewer’s suggestion, we have added a brief subsection titled “Patient and Public Involvement” at the end of the Methods section to explicitly state this (lines 214-217).

Clarification of Subsection on Baseline Characteristics: We appreciate the reviewer’s request for clarification. The first subsection indeed describes the baseline characteristics of the study participants. We have revised the subsection heading and introductory sentence to explicitly reflect that it presents baseline demographic and clinical characteristics at admission (line 192).

Where is each ethnicity group mentioned? If provided in Table 1, this information will help better grasp the fact that the Dagomba people are the lowest-risk group (with the highest survival probability) in the studied population of Ghana

We thank the reviewer for this comment. Ethnicity is reported in Table 1 under baseline participant characteristics, as well as described in the study variables section.

Did the participants report at baseline a family history of major adverse cardiovascular events other than stroke among their first-degree relatives?

We thank the reviewer for this comment. At baseline, participants reported only a family history of stroke among their first-degree relatives. Information on other major adverse cardiovascular events was not collected in this study. They only provided comorbidities, including hypertension, and so on, that they were suffering from by themselves.

What were the frequencies of stroke at baseline? What are the frequencies of haemorrhagic and ischaemic stroke at baseline as well as the arterial territories in ischaemic stroke patients and the cerebral localizations of haemorrhagic stroke?

We thank the reviewer for this comment. The frequencies of stroke subtypes at baseline are reported in lines 173–174 and summarized in Table 1, with ischaemic stroke accounting for 51.7% and haemorrhagic stroke for 48.3% of cases. Data on arterial territories in ischaemic stroke patients and cerebral localizations of haemorrhagic stroke were not collected in this study and therefore cannot be reported

Reviewer #1

The title needs to be rewritten We revised the title to clarify the outcome and study design. Suggested title: “In-hospital survival and predictors of mortality among stroke patients at a Tertiary Hospital in Ghana: a retrospective cohort study.”

Summarise the abstract introduction We appreciate the reviewer’s suggestion to make the abstract more concise. In response, we have summarised the Introduction section of the abstract to focus on the essential background and study objective. The revised text now reads:

“Stroke is a leading cause of death and disability globally and imposes a growing burden on Ghana’s healthcare system. This study aimed to estimate in-hospital survival rates and identify predictors of mortality among stroke patients admitted to a tertiary hospital in Ghana over a 3-year period.”

Your literature review is weak. Please cite examples of similar studies. We appreciate the reviewer’s feedback. In response, we have strengthened the literature review by including additional studies on stroke survival and its determinants. Specifically, we have now cited:

• Sarfo FS, et al. Long-Term Outcomes of Stroke in a Ghanaian Outpatient Clinic. J Stroke Cerebrovasc Dis. 2018.

• Sarfo FS, et al. Key determinants of long-term post-stroke mortality in Ghana.(line 86)

Provide logical reasoning for all comparisons (e.g., why your median survival differs from other studies). We rewrote the comparison paragraph to state explicitly why median survival may differ across studies: This difference is likely explained by contextual factors such as variation in stroke severity and complication rates, health systems, and variation in sample size (lines 302-303)

Reviewer #2

1. The follow-up period (mean survival: 67 days) is not mentioned. Thank you for pointing this out. We have revised the Methods section of the abstract to include the mean follow-up duration (Line 32). It now reads: “Kaplan–Meier survival analysis was used to estimate the survival rate of stroke patients (mean follow-up: 67 days).”

2. A protective effect of diabetes is reported as an unexpected finding, but it is presented without sufficient explanation or cautious interpretation. We appreciate this important comment. In the revised abstract, we have reported the association between diabetes mellitus and mortality more cautiously, noting that the apparent lower risk may be due to differences in case management or residual confounding (lines 45 – 48). The Results section now states: “Diabetes mellitus showed an apparently lower risk of mortality (AHR = 0.62, 95% CI: 0.47–0.82); this finding should be interpreted cautiously as it may reflect differences in case management or residual confounding.” (lines 280-282)

The definition of the outcome is unclear. The text indicates that “survival status” was determined at discharge (alive or dead), but the Kaplan–Meier results report a mean survival of 67 days with follow-up up to 50 days. This creates confusion as to whether the authors considered in-hospital survival only or included post-discharge follow-up. Suggestion: Authors should explicitly clarify how the outcome was defined (discharge status only or extended follow-up). Without this, it is difficult to compare results with other studies and to interpret the Kaplan–Meier curves. Thank you for this important observation. We have revised the Materials and Methods section to explicitly state that survival status was determined at hospital discharge only; no post-discharge follow-up was included. The time-to-event variable used for Kaplan–Meier analysis was calculated as the number of days from admission to discharge or in-hospital death. This clarification has been added to the “Study Variables” (lines 141-144) and “Data Analysis” (lines 175-176) subsections.

3. The follow-up duration and key study limitations (e.g., retrospective design) shoul

---

## [Decision Letter · Decision Letter 2]

18 Dec 2025

PONE-D-25-39162R2

IN-HOSPITAL SURVIVAL AND PREDICTORS OF MORTALITY AMONG STROKE PATIENTS AT A TERTIARY HOSPITAL IN GHANA: A RETROSPECTIVE COHORT STUDY

PLOS One

Dear Dr. Abdulai,

Thank you for submitting your manuscript to PLOS ONE. After careful consideration, we feel that it has merit but does not fully meet PLOS ONE’s publication criteria as it currently stands. Therefore, we invite you to submit a revised version of the manuscript that addresses the points raised during the review process.

We look forward to receiving your revised manuscript.

Kind regards,

Mickael Essouma, M. D.

Academic Editor

PLOS OneJournal

Requirements:

Additional Editor Comments:

The reviewer has recommended manuscript acceptance for publication. However, there are still some issues that need to be addressed before the manuscript can be finally accepted.

First, it is still unclear whether you assessed predictors of survival or predictors of death given the multiple inconsistencies between the manuscript's title and different parts of the manuscript (eg, text of the methos and results sections and titles of tables 4, 5) in this regard. Consider making that information clear from the title of the manuscript to the end of the manuscript.

Second, you did not specify the study design in the materials and methods section whilst you clearly specified the design and provided a valid reason why this study was a predictive retrospective cohort. You are therefore urged to copy that information from the response letter and paste it in the first sub-section of the Material and Methods section which would be termed "Study design".

Third, in the response letter, you claimed that you could not stratify data by ischaemic and hemorrhagic stroke subtypes. However, you did not mention that information in the limitations statement of the Discussion section. Consider addressing this issue in the manuscript per se.

Mickael Essouma, M.D.

Reviewers' comments:

Reviewer's Responses to Questions

**Comments to the Author**

1. If the authors have adequately addressed your comments raised in a previous round of review and you feel that this manuscript is now acceptable for publication, you may indicate that here to bypass the “Comments to the Author” section, enter your conflict of interest statement in the “Confidential to Editor” section, and submit your "Accept" recommendation.

Reviewer #1: (No Response)

2. Is the manuscript technically sound, and do the data support the conclusions?

Reviewer #1: (No Response)

3. Has the statistical analysis been performed appropriately and rigorously? 

Reviewer #1: (No Response)

4. Have the authors made all data underlying the findings in their manuscript fully available?

Reviewer #1: (No Response)

5. Is the manuscript presented in an intelligible fashion and written in standard English?

Reviewer #1: (No Response)

6. Review Comments to the Author

Reviewer #1: (No Response)

7. PLOS authors have the option to publish the peer review history of their article (what does this mean?). If published, this will include your full peer review and any attached files.

Reviewer #1: No

---

## [Author Response · Author response to Decision Letter 3]

20 Dec 2025

First, it is still unclear whether you assessed predictors of survival or predictors of death given the multiple inconsistencies between the manuscript's title and different parts of the manuscript (eg, text of the methods and results sections and titles of tables 4, 5) in this regard. Consider making that information clear from the title of the manuscript to the end of the manuscript. We have revised the manuscript to consistently focus on predictors of in-hospital mortality. The title, methods, results, and Tables 1 and 2 have been updated accordingly

Second, you did not specify the study design in the materials and methods section whilst you clearly specified the design and provided a valid reason why this study was a predictive retrospective cohort. You are therefore urged to copy that information from the response letter and paste it in the first sub-section of the Material and Methods section which would be termed "Study design" We thank the Academic Editor for this important observation. We acknowledge that although the study design was clearly described in our previous response letter, it was not explicitly stated within the Materials and Methods section of the manuscript. We have now addressed this by adding a dedicated subsection titled “Study design” as the first subsection of the Materials and Methods section. This subsection clearly states that the study employed a retrospective cohort design and provides the rationale for its appropriateness. The relevant text has been incorporated into the manuscript (Page 6, Lines 101–106).

Third, in the response letter, you claimed that you could not stratify data by ischaemic and hemorrhagic stroke subtypes. However, you did not mention that information in the limitations statement of the Discussion section. Consider addressing this issue in the manuscript per se. We have clarified this issue in the Discussion section by explicitly stating that although stroke subtypes were reported at baseline, stratified analyses by ischaemic and haemorrhagic stroke were not performed, and detailed localization data were unavailable (Page 26, Lines 390–395).

---

## [Editor Report · Decision Letter 3]

22 Dec 2025

IN-HOSPITAL SURVIVAL AND PREDICTORS OF MORTALITY AMONG STROKE PATIENTS AT A TERTIARY HOSPITAL IN GHANA: A RETROSPECTIVE COHORT STUDY

PONE-D-25-39162R3

Dear Dr. Abdulai,

We’re pleased to inform you that your manuscript has been judged scientifically suitable for publication and will be formally accepted for publication once it meets all outstanding technical requirements. Congratulations!

Kind regards,

Mickael Essouma, M. D.

Academic Editor

PLOS One

Additional Editor Comments (optional):

To avoid disrupting the flow of the results section, consider moving the «Patient and Public Involvement» sub-section, which is currently in the results section (lines 220-223), to the end of the Materials and Methods section.
---

## [Editor Report · Acceptance letter]

PONE-D-25-39162R3

PLOS One

Dear Dr. Abdulai,

I'm pleased to inform you that your manuscript has been deemed suitable for publication in PLOS One. Congratulations! Your manuscript is now being handed over to our production team.

Kind regards,

on behalf of

Dr. Mickael Essouma

Academic Editor

PLOS One